# Genome-Wide Pleiotropy Study Identifies Association of *PDGFB* with Age-Related Macular Degeneration and COVID-19 Infection Outcomes

**DOI:** 10.3390/jcm12010109

**Published:** 2022-12-23

**Authors:** Jaeyoon Chung, Viha Vig, Xinyu Sun, Xudong Han, George T. O’Connor, Xuejing Chen, Margaret M. DeAngelis, Lindsay A. Farrer, Manju L. Subramanian

**Affiliations:** 1Department of Medicine (Biomedical Genetics), Boston University Chobanian & Avedisian School of Medicine, Boston, MA 02118, USA; 2Department of Ophthalmology, Boston University Chobanian & Avedisian School of Medicine, Boston, MA 02118, USA; 3Department of Medicine (Pulmonary & Critical Care), Boston University Chobanian & Avedisian School of Medicine, Boston, MA 02118, USA; 4Department of Population Health Sciences and Department of Ophthalmology and Visual Sciences, University of Utah School of Medicine, Salt Lake City, UT 84112, USA; 5Department of Ophthalmology, Jacobs School of Medicine and Biomedical Sciences, State University of New York at Buffalo and VA Research Service, Veterans Affairs Western New York Healthcare System, Buffalo, NY 14203, USA; 6Department of Neurology, Boston University Chobanian & Avedisian School of Medicine, Boston, MA 02118, USA; 7Departments of Epidemiology and Biostatistics, Boston University School of Public Health, Boston, MA 02118, USA

**Keywords:** GWAS, AMD, COVID-19, Mendelian randomization, PDGFB

## Abstract

Age-related macular degeneration (AMD) has been implicated as a risk factor for severe consequences from COVID-19. We evaluated the genetic architecture shared between AMD and COVID-19 (critical illness, hospitalization, and infections) using analyses of genetic correlations and pleiotropy (i.e., cross-phenotype meta-analysis) of AMD (*n* = 33,976) and COVID-19 (*n* ≥ 1,388,342) and subsequent analyses including expression quantitative trait locus (eQTL), differential gene expression, and Mendelian randomization (MR). We observed a significant genetic correlation between AMD and COVID-19 infection (r_G_ = 0.10, *p* = 0.02) and identified novel genome-wide significant associations near *PDGFB* (best SNP: rs130651; *p* = 2.4 × 10^−8^) in the pleiotropy analysis of the two diseases. The disease-risk allele of rs130651 was significantly associated with increased gene expression levels of *PDGFB* in multiple tissues (best eQTL *p* = 1.8 × 10^−11^ in whole blood) and immune cells (best eQTL *p* = 7.1 × 10^−20^ in T-cells). PDGFB expression was observed to be higher in AMD cases than AMD controls {fold change (FC) = 1.02; *p* = 0.067}, as well as in the peak COVID-19 symptom stage (11–20 days after the symptom onset) compared to early/progressive stage (0–10 days) among COVID-19 patients over age 40 (FC = 2.17; *p* = 0.03) and age 50 (FC = 2.15; *p* = 0.04). Our MR analysis found that the liability of AMD risk derived from complement system dysfunction {OR (95% CI); hospitalization = 1.02 (1.01–1.03), infection = 1.02 (1.01–1.03) and increased levels of serum cytokine PDGF-BB {β (95% CI); critical illness = 0.07 (0.02–0.11)} are significantly associated with COVID-19 outcomes. Our study demonstrated that the liability of AMD is associated with an increased risk of COVID-19, and PDGFB may be responsible for the severe COVID-19 outcomes among AMD patients.

## 1. Introduction

Severe acute respiratory syndrome coronavirus 2 (SARS-CoV-2) infection is the cause of coronavirus disease 2019 (COVID-19), which has become a global health crisis [1]. Although much of the disease burden relates to respiratory failure and sepsis, clinical sequelae and disease severity after COVID-19 infection are highly variable [2]. A systematic review of comorbidities associated with the risk of severe COVID-19 outcomes determined that chronic conditions such as obesity, hypertension, cardiovascular disease, cerebrovascular disease, respiratory disease, kidney disease, and malignancy are clinical risk factors for fatal or severe COVID-19 outcomes and infection risk [3].

Recently, evidence has emerged to suggest that age-related macular generation (AMD) is also a clinical risk factor for increased risk of infection [4] and mortality [5,6] from COVID-19. AMD, the leading cause of irreversible vision loss in Western countries for people over age 50, is a chronic age-related neurodegenerative disease of the retina and retinal pigment epithelium (RPE) associated with inflammation and aging [7,8]. AMD is a co-morbid condition that has been reported to confer higher risk of severe complications of SARS-CoV-2 infection, including respiratory failure and death (25%), a risk which is higher than Type 2 diabetes (21%) and obesity (13%) [5,6].

Considering these observations, we hypothesized that AMD and COVID-19 share common genetic risk factors. In this study, we evaluated the genetic correlation of AMD and COVID-19 and attempted to identify the genetic determinants underlying shared mechanisms of these two diseases using cross-phenotype meta-analysis (i.e., pleiotropy analysis). The relevance of AMD risk and COVID-19 (infection and severity) to the top-ranked finding was further examined by analyses of transcriptome datasets of AMD and COVID-19 and Mendelian randomization.

## 2. Methods

### 2.1. Genome-Wide Association Summary Statistics for AMD and COVID-19

We obtained summary statistics from an AMD GWAS conducted by the International Age-related Macular Degeneration Genomics Consortium (IAMDGC), which included 16,144 AMD cases and 17,832 controls of European ancestry samples across 26 datasets [9]. Different subtypes of AMD were analyzed together with summary-level data in the IAMDGC GWAS, including geographic atrophy (dry type), choroidal neovascular membrane (wet type), and mixed AMD [9].

Next, we used the COVID-19 Host Genetics Initiative website, round 5 (released on 18 January 2021; https://www.covid19hg.org) to obtain European population-based GWAS summary statistics for three different COVID-19-related outcomes in a European population—(A) *critical illness*, defined as those patients on respiratory support or dying from COVID-19 (5101 cases and 1,383,241 controls), (B) *hospitalization* due to severe or moderate COVID-19 symptoms (9986 cases and 1,877,672 controls), and (C) *infection rates* with SARS-CoV-2, including testing positive for both symptomatic and asymptomatic cases—against unscreened population controls (38,984 cases and 1,644,784 controls) (Appendix A) [10].

### 2.2. Genetic Correlation

Genetic correlations of AMD with the three COVID-19 outcomes were calculated by linkage disequilibrium score (LDSC) [11] (https://github.com/bulik/ldsc) and GNOVA [12] (https://github.com/xtonyjiang/GNOVA). Genetic correlation (r_G_) estimates outside of (−1, 1) were considered as −1 or 1, respectively.

### 2.3. Pleiotropy Genome-Wide Association Study of AMD and COVID-19

We conducted a cross-phenotype meta-analysis to identify genetic factors shared between AMD and COVID-19 outcomes (critical illness, hospitalizations, and infections) using the Multi-Trait Analysis in GWAS (MTAG) software [13]. We performed separate MTAG analyses for AMD with each of three COVID-19 outcomes (critical illness, hospitalizations, and infections). An SNP was considered to have a pleiotropic effect on two traits when the *p*-value for the test statistic from the cross-phenotype analysis was at least one order of magnitude more significant than the *p*-values for both traits and the SNP was at least nominally significant (*p* < 0.05) with each of the individual traits [13,14].

### 2.4. Gene-Based Association Analysis

Using expression data (blood for AMD; blood, lung, aorta, and coronary artery for COVID-19) from PredictDB (http://predictdb.org), we applied a gene-based association method implemented in S-PrediXcan [15]. A genome-wide significance threshold was set as α = 0.05 with Bonferroni correction for the maximum number of genes tested (8500) among the tissues (*p* < 5.88 × 10^−6^ = 0.05/8500).

### 2.5. Expression/DNA Methylation Quantitative Trait Loci and Colocalization Analyses

The top-ranked SNPs identified in the MTAG analysis were examined using publicly available expression quantitative trait loci (eQTL) databases, including the GTEx Portal [16] (https://www.gtexportal.org/) and FIVEx [17] (https://fivex.sph.umich.edu/), which provides summary statistics for tissue-specific associations between allele dosages of SNPs and transcription-level expression. Colocalization analysis was performed to determine whether the eQTL associations between the significant SNPs (identified by MTAG) and candidate genes (identified by *cis*-eQTL analysis) are explained (i.e., colocalized) by the associations between the same SNPs and the disease using the coloc package [18] with the eQTL summary statistics from GTEx. We selected the window of colocalization to be 500 kilobase (kb) pairs spanning the most significant SNPs. A posterior probability (PP) exceeding 90% was used as evidence of colocalization. We searched for DNA methylation sites that were strongly associated (*p* < 1 × 10^−5^) with the top-ranked SNPs identified in the GWAS using the DNA methylation QTL (mQTL) consortium database (GoDMC; www.godmc.org.uk) [19], which provides results for association tests of SNPs with blood DNA methylated sites derived from meta-analysis of 36 genome-wide DNA methylation studies (N = 27,750).

### 2.6. Differential Gene Expression Analysis

To understand the disease relevance of the top findings from our pleiotropy analysis, differential gene expression between AMD cases and controls was evaluated using microarray gene expression data derived from human RPE/choroid tissues obtained from 9 AMD cases and 6 age-matched healthy controls downloaded from the Gene Expression Omnibus (GEO; accession ID: GSE50195) database [20].

In addition, we obtained RNA sequencing (RNA-Seq) data from GEO (GSE161731), derived from 96 peripheral blood samples from 32 COVID-19 patients and 8 controls at multiple time points corresponding to different disease stages measured from symptom onset: 19 samples in the early/progressive stage (within 10 days); 36 samples in the middle stage (between 11 and 20 days), which shows the peak in COVID-19 symptoms; 22 samples in the late stage (>21 days) [21]. Twelve COVID-19 patients were hospitalized (6 of which were in the early stage and 6 in the middle stage, with none in the late stage). The controls in this group {mean age (SD) = 18.4 (0.6) years} were younger than the COVID-19 patients {mean ages: early = 46.4 (19.7), middle = 41.1 (16.4), late = 41.8 (13.4)} (Appendix A). Additional demographic information for subjects in these datasets for AMD and COVID-19 is summarized in Appendix A.

The association of log_2_-transformed values of gene expression with AMD status was evaluated with linear regression models including covariates for age, sex, and batch. Mean levels of gene expression were compared between COVID-19 patients and controls and across disease stages using ANOVA. Because the COVID-19-free controls {mean age (SD) = 18.37 (0.60)} were more than 20 years younger than the COVID-19 patients {mean age (SD) = 42.55 (16.38)} and our interest is genetic factors shared between COVID-19 and AMD, which usually manifests after age 50 [22], we also evaluated gene expression differences between COVID-19 stages in different age groups defined using age cutoffs of >30 years, >40 years, and >50 years. The association of gene expression with binary disease stage outcomes (early vs. middle, early vs. middle/late) was tested using generalized estimating equation regression models including covariates for age, sex, batch, and hospitalization (yes/no).

### 2.7. Pathway Analysis Using Gene Ontology

To identify mechanisms shared between the two diseases, we conducted pathway analysis based on gene ontology (GO) categories using MAGMA software [23] seeded with the genes in regions of associated SNPs (*p* < 10^−5^) identified in the pleiotropy analysis. A total of 5917 GO categories were tested, and a *p*-value threshold was set to 8.45 × 10^−6^ after applying the Bonferroni correction for multiple comparisons (0.05/5917).

### 2.8. Mendelian Randomization Analysis

We employed Mendelian randomization (MR) to investigate the causal relationships between AMD as an exposure and the three COVID-19 traits as outcomes using three approaches. First, we tested SNPs in all genes previously associated with AMD at a genome-wide significance level (*p* < 5.0 × 10^−8^) [9] as instrument variables (IVs). Next, we focused on causal relationships between the AMD risk in the complement system and the COVID-19 phenotypes using IV SNPs located in or near several genes in the complement system (*C2*/*C4A*/*C4B*/*CFB*, C3, *CFH*/*CFHR3*, *CFI*, and *VTN*), which is the most relevant pathway for AMD [24,25]. Finally, because *PDGFB* emerged from our MTAG analysis as significantly associated with the joint outcomes of AMD and COVID-19, we tested the causal relationship between the serum concentration of PDGF-BB (which is the homodimer form of the cytokine PDGF-B) and the COVID-19 phenotypes. To do so, we used the summary-level statistics for a previous genome-wide QTL study for the serum concentration of PDGF-BB [26] as the exposure.

For all analyses, instrument SNPs were pruned by linkage disequilibrium (LD; r^2^ < 0.001) based on the European 1000 Genomes reference and were further excluded if they had palindromic alleles (A/T or C/G), a minor allele frequency < 5%, or imputation quality (R^2^) < 0.8. In addition, instrument SNPs that were significantly associated with one of the three COVID-19 phenotypes (*p* < 0.05) were excluded to avoid bias from the horizontal pleiotropic effect. Twenty-three independent SNPs associated with AMD risk were selected as instrument variables for analyses including AMD as the exposure (Appendix A). Analyses focused on AMD resulting from complement system dysfunction utilized the following complement pathway gene SNPs as IVs: rs429608 for *C2*/*C4A*/*C4B*/*CFB*, rs11569415 for *C3*, rs10801558 for *CFH*/*CFHR3*, rs1003390 for *CFI*, and rs1108005 for *VTN*. MR analyses of PDGF-BB serum concentration as the exposure included rs72777070, rs13412535, rs2324229, or rs4965869 as the IV (Appendix A). We applied the random-effect two-sample inverse variance weighted (IVW) method under the assumption that all SNPs are valid instruments for a specific exposure using Mendelian randomization [27]. We also conducted sensitivity analyses with pleiotropy-robust two-sample MR methods, including weighted median MR and MR-Egger, to compare the MR estimates between the MR models [27]. The intercept of the MR-Egger model represents a test for directional pleiotropy.

## 3. Results

### 3.1. Genetic Correlation of AMD with COVID-19 Outcomes

LDSC analysis showed that the three COVID-19 outcomes are perfectly genetically correlated (r_G_s = 1.0, *p* < 3.3 × 10^−13^), indicating that the same genetic factors largely influence these traits. However, the correlations estimated using GNOVA were somewhat lower (0.51 < r_G_s < 0.84, *p* < 1.2 × 10^−21^). There was a modest genetic correlation between AMD and COVID-19 infection rate (LDSC r_G_ = 0.11, *p* = 0.07; GNOVA r_G_ = 0.10, *p* = 0.02; Figure 1) but not with critical illness and hospitalizations, perhaps because the sample size of COVID-19 infections was at least four times larger than the sample of those with severe illness and hospitalization.

### 3.2. Multi-Trait Analysis of GWAS for AMD and COVID-19

There was modest evidence for genomic inflations with each of the three COVID-19 outcomes in the MTAG results for AMD (Appendix A; 1.07 < λ ≤ 1.08). Genome-wide significant (GWS; *p* < 5.0 × 10^−8^) associations were observed at 60 loci in the MTAG results for AMD with the three COVID-19 outcomes (Appendix A; Appendix A). Notably, 49 of these loci were previously identified as associated with AMD [9] and 10 other loci were previously associated with one of the COVID-19 outcomes [10]. In addition to these previously identified loci, we identified GWS association with two SNPs located 3.2 kb pairs upstream of a novel locus, *PDGFB* (rs130651, MTAG *p* = 2.44 × 10^−8^ and rs4820371, MTAG *p* = 2.60 × 10^−8^; Table 1; Figure 2A). Both SNPs showed evidence suggestive of association with AMD when analyzed separately (rs130651, *p* = 1.36 × 10^−7^ and rs4820371, *p* = 1.71 × 10^−7^) and were marginally associated with COVID-19 infection (rs130651, *p* = 0.024 and rs4820371, *p* = 0.003) (Table 1). In addition, there was GWS evidence of pleiotropy with both SNPs when combining the GWAS results for AMD with those for critical illness (rs130651, *p* = 3.5 × 10^−8^ and rs4820371, *p* = 6.1 × 10^−8^) and hospitalization (rs130651, *p* = 4.4 × 10^−8^ and rs4820371, *p* = 5.7 × 10^−8^), but neither SNP was associated with critical illness (*p* > 0.27) or hospitalization (*p* > 0.87; Table 1). Gene-based testing did not identify associations with any novel genes after multiple testing correction (*p* < 5.88 × 10^−6^; Appendix A).

### 3.3. PDGFB SNP rs130651 Regulates PDGFB Expression

Examination of the association of the two GWS SNPs identified in the MTAG analysis (rs130651 and rs4820371) with expression in blood of 27 genes located within 500 kb of these variants revealed that rs30651 is a study-wide significant *cis*-eQTL for *PDGFB* in blood in the GTEx database (*p* = 1.8 × 10^−11^; Figure 2B) and in data from Lepik et al. (*p* = 8.15 × 10^−21^) and Schmeidel et al. (*p* = 3.8 × 10^−9^) reported in the FIVEx database (Table 2). Specifically, the disease risk rs130651-*A* allele was significantly associated with increased expression of *PDGFB.* Rs4820371 was significantly associated with expression of *PDGFB* in blood, but only in the Lepik et al. study (*p* = 5.78 × 10^−8^). Rs130651 was also significantly associated with *PDGFB* expression in several immune cell types, most notably T-cells (*p* = 7.1 × 10^−20^; Table 2; Appendix A). The evidence supporting either SNP as a *cis*-eQTL with nearby gene *CACNA1I* was markedly weaker (Appendix A). Colocalization analysis of 1761 SNPs in the *PDGFB* locus demonstrated that the genetic association signal at *PDGFB* colocalized with the *PDGFB* eQTL signals in blood (PP = 93.8%).

According to the mQTL GoDMC database, the disease risk alleles of rs130651(*A*) and rs4820371 (*T*) are significantly associated with decreased DNA methylation levels of CpG sites located 70–145 kb upstream of *PDGFB* (Appendix A). The most significant *cis*-mQTL association was observed between rs130651 and cg11247378, which is located 143.9 kb from *PDGFB* (β = −0.18; *p* = 9.74 × 10^−64^).

### 3.4. PDGFB Is Differentially Expressed in AMD and COVID-19

*PDGFB* expression in human RPE/choroid tissue was slightly higher in nine AMD cases compared to six controls, although the association was not statistically significant {fold change (FC) = 1.02, *p* = 0.065; Figure 2C}. There was no difference in *PDGFB* expression in healthy controls and the three COVID-19 stages (ANOVA *p*-value = 0.85; Appendix A). However, *PDGFB* expression was significantly higher among COVID-19 patients in the middle stage compared to those in the early stage who were older than age 40 (FC = 2.17, *p* = 0.03; Figure 2D) or older than age 50 (FC = 2.15, *p* = 0.04), but not in the total group of COVID-19 patients in these stages (FC = 0.99, *p* = 0.61; Appendix A and Appendix A) or among early-stage patients compared to the middle- and late-stage patients combined (*p* > 0.10; Appendix A).

### 3.5. Pathway Analysis

Pathway analysis including genes located at or near top-ranked SNPs in any of the MTAG analyses revealed four significant (*p* < 8.45 × 10^−6^) GO categories, including high density lipoprotein particle remodeling (*p* = 1.71 × 10^−8^), reverse cholesterol transport (*p* = 1.53 × 10^−6^), humoral immune response mediated by circulating immunoglobulin (*p* = 4.31 × 10^−6^), and extracellular structure organization (*p* = 3.53 × 10^−6^), which includes *PDGFB* (Appendix A).

### 3.6. Mendelian Randomization

No significant causal relationships between the AMD genome-wide risk and any of the COVID-19 outcomes were observed (*p* > 0.31; Figure 3). However, AMD likely arising from genetic predisposition to dysfunction in the complement system was significantly associated with increased risk for COVID-19 hospitalization (OR = 1.02 [1.01–1.03], *p* = 4.02 × 10^−4^) and infection {OR = 1.02 (1.01–1.03), *p* = 1.34 × 10^−3^}. In addition, the serum PDGF-BB concentration was significantly associated with increased risk of COVID-19 critical illness {β = 0.07 (0.02–0.11), *p* = 4.00 × 10^−3^}. Sensitivity analysis showed that none of these results were confounded by directional pleiotropy (Appendix A).

## 4. Discussion

Recent studies have shown an increased risk of infection and mortality from COVID-19 among AMD patients [4,5,6]. We hypothesized that biological mechanisms shared between the two diseases exist and attempted to identify underlying genetic factors shared between them using data from GWAS for AMD and three outcomes related to COVID-19. Analysis of these data using a pleiotropy model revealed one novel genome-wide significant association in the *PDGFB* locus for AMD and COVID-19 infection.

Considering variation across the genome as a whole, we observed a modest genetic correlation between AMD and COVID-19 infection, which indicates that some genetic factors contribute to risk for both diseases. We did not observe a strong genetic correlation of AMD with COVID-19 critical illness and hospitalization. However, since all three COVID-19 outcomes share strongly genetically correlated influence, it is likely that the sevenfold smaller sample for critical illness and fourfold smaller sample for hospitalization compared to infection rate limited our ability to detect significant genetic correlations between AMD and these two COVID-19 outcomes. Results from MR analysis including only SNPs in the genes for the complement system suggest a causal relationship between complement pathway dysfunction in AMD and subsequent infection by and hospitalization due to COVID-19. In contrast, considering all genes associated with AMD, there was no evidence for causality of AMD on COVID-19 outcomes. Taken together, these MR findings may explain the modest genetic correlations between the two diseases when comparing entire genomes, yet they reinforce the clinical observation of higher risk for infection by and severe illness from COVID-19 among AMD patients [5,6].

We evaluated various omics data to identify possible functional links between *PDGFB* and AMD and COVID-19 outcomes. The top-ranked SNP from the pleiotropy analysis of AMD and COVID-19 infection risk, rs130651, is a significant *cis*-acting eQTL and mQTL, in which the disease risk allele *A* is associated with higher *PDGFB* expression and lower DNA methylation levels in the *PDGFB* promoter region. In addition, colocalization analysis confirmed that, among the 28 genes located under the association peak, *PDGFB* is most likely to be functionally linked to AMD and COVID-19 infection. Taken together, the findings from our pleiotropy and expression/methylation QTL analyses suggest that increased *PDGFB* expression is associated with increased risk for AMD and COVID-19 infection and severity.

Our DGE analysis of the AMD sample showed that *PDGFB* tended to be more highly expressed in AMD cases compared to controls in human RPE tissue, although the result was not statistically significant (*p* = 0.067), perhaps due to the small sample size. However, a recent snRNA-Seq expression study in human retina/RPE tissues showed that *PDGFB* expression was significantly higher in AMD cases compared to controls, particularly in myeloid immune cells (FC = 6.12, *p* = 4.89 × 10^−9^) [28]. Myeloid cells, together with RPE and glia, have been considered to play important roles in the etiology of AMD [28]. Our expression findings are also consistent with a previous study showing that *PDGFB* expression in blood was significantly higher in neovascular (wet type) AMD patients than in AMD-free controls, and significantly correlated with the expression of vascular endothelial growth factor (VEGF) [29].

Our finding of significantly higher *PDGFB* expression among COVID-19 patients ages 40 years and older in the middle stage (11–21 days after the symptom onset) compared to the early stage (0–10 days) is consistent with the observation that COVID-19 symptoms peak 10 days after symptom onset [30]. Although we did not observe significantly different *PDGFB* expression between COVID-19 patients and COVID-19-free controls, recent studies showed significantly higher serum PDGF-BB concentration in COVID-19 patients compared to the controls (Appendix A) [30,31]. Furthermore, Liu et al. found that increased serum PDGF-BB concentrations are significantly associated with increased viral load (*p* = 0.002) and lung injury (*p* = 0.001) in COVID-19 patients [32]. These collective findings suggest that lowering *PDGFB* expression and serum PDGF-BB concentration may reduce the severity of COVID-19, particularly among older people.

*PDGFB* encodes a member of the protein family comprised of both platelet-derived growth factors (PDGF). PDGF proteins bind and activate PDGC-receptor tyrosine kinases, which have a role in a wide range of developmental processes. *PDGFB* is involved in the recruitment of pericytes to newly formed blood vessels in the course of physiological angiogenesis during retinal development, as well as in the context of pathological neovascularization that occurs in diabetic retinopathy and AMD [33]. In addition, blocking PDGF signaling is thought to inhibit the pericyte–endothelial process in neovascularization [34]. Therapeutic strategies combining anti-VEGF therapy with antagonists for blocking PDGF signaling have been considered even more effective than the single VEGF treatment and are currently under investigation (e.g., phase III trials—https://clinicaltrials.gov/ identifiers: NCT01944839, NCT01940900, and NCT01940887) [33,35].

A growing body of evidence suggests that the deleterious impact of the exaggerated immune response in COVID-19 patients is a consequence of the SARS-CoV-2 virus activating and exploiting multiple complement cascades, which drive inflammation and thrombosis and result in severe clinical outcomes and poor prognosis [36,37,38]. Interestingly, the Beaver Dam Eye Study [39] and a retrospective cohort study of data in the National Health Insurance Database of Taiwan [40] discovered that subjects with signs and symptoms of obstructive pulmonary diseases such as emphysema and chronic obstructive pulmonary disease had a higher incidence of AMD, presumably due to inflammatory dysregulation. Several AMD studies have identified genetic variants associated with complement dysfunction and its impact on inflammation, oxidative stress, and metabolism, and these pathways have been found to be significantly upregulated and dysfunctional in COVID-19 infection as well [24,25,38,41,42,43,44].

Our study has limitations. First, the patients in the COVID-19 infection cohort were based on self-reporting or confirmed patients at clinics, which indicates that asymptomatic COVID-19 patients may not be included. This may have contributed to the underpowering of the analysis, further confirmed by the imbalance in sample size between AMD and COVID-19 outcomes for critical illness and hospitalization. Second, the unscreened population controls may have included patients with both asymptomatic and symptomatic disease who were unable to be tested or obtain care during the height of the pandemic. A third concern is the lack of genetic correlation between AMD and the COVID-19 outcomes of critical illness and hospitalization. This may be explained in part by the ascertainment of the AMD and COVID-19 GWAS cohorts included in this study. The AMD sample was comprised primarily of older individuals (mean age = 73.7 ± 9.4 years) whereas the COVID-19 sample included persons across a wide age spectrum (mean age ranges from 36 different datasets = 35.6–72.3 years). Genetic correlation in the overall sample would be weakened if it is age-dependent. Furthermore, a study of a large Korean cohort (N = 135,435) showed that only the exudative (wet) form of AMD was significantly associated with severe COVID-19 illness and COVID-19 infection [6]. In addition, Allegrini et al. found a significant loss in visual function among the exudative AMD patients who experienced COVID-19 infection compared to AMD patients before the pandemic [45]. Unfortunately, we were unable to evaluate genetic correlations between COVID-19 outcomes and exudative and non-exudative AMD separately because our analyses were conducted using AMD GWAS summary-level data. Moreover, because the distribution of AMD subtypes in the GWAS dataset is highly imbalanced (67.4% exudative, 19.7% geographic atrophy, 12.9% with each form in one eye), separate analyses for each AMD subtype would not be meaningful. A balanced age distribution between the AMD and COVID-19 cohorts and AMD subtypes might have resulted in a different estimate of the genetic correlation between the two diseases. Although age differences between the individuals in the AMD and COVID-19 datasets may have influenced the genetic correlation between the two disorders, age disparity does not impact the results of pleiotropy analyses, which are conducted separately within datasets for each disease and subsequently pooled by meta-analysis. This conclusion is supported by pleiotropy studies for diseases with very distinct onset age distributions, such as childhood obesity and adult cardiometabolic diseases [46] and autism spectrum disorder and schizophrenia [47]. Finally, our study was conducted using GWAS and gene expression datasets obtained from European ancestry samples, which limits the generalizability of our results to other ancestral groups. Future studies in other populations are needed to replicate and extend our findings.

In summary, our findings add to the body of evidence for an increased risk of infection and mortality from COVID-19 among AMD patients. Our analysis lends credence to previously reported clinical studies that found those with AMD are an at-risk group for COVID-19 infection and severe disease, and that this increased risk may have a genetic basis. The discovery of shared genetic risk factors via GWAS will require a larger sample size for critical illness and hospitalizations to better understand the shared pathology and risk factors that contribute to worsening clinical outcomes in both disease states.

## Figures and Tables

**Figure 1 jcm-12-00109-f001:**
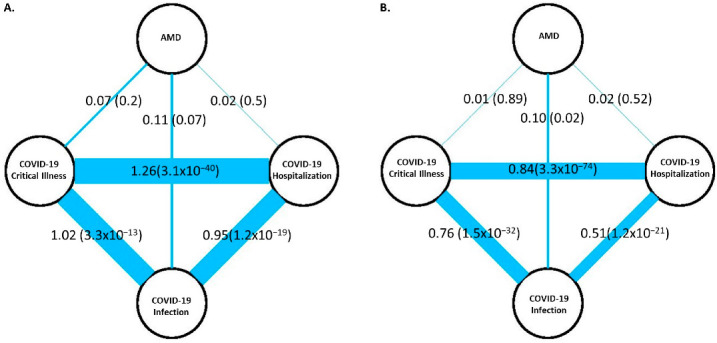
Genetic correlations between AMD and three COVID-19 outcomes, including critical illness, hospitalization, and infections, which were computed by (**A**) LDSC and (**B**) GNOVA. Line width is proportional to genetic correlations between the four phenotypes. The numbers show the genetic correlations and *p*-values amongst the phenotypes.

**Figure 2 jcm-12-00109-f002:**
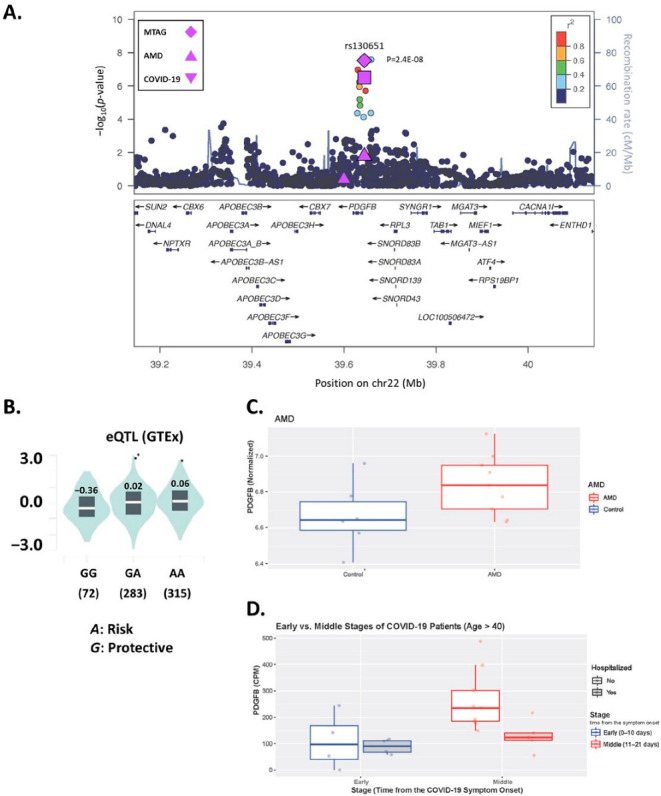
(**A**) Regional association plot for the genome-wide significant associations in the MTAG analysis of AMD and COVID-19 infections. Rs130651 is the most significant SNP in this locus (MTAG *p*-value = 2.44 × 10^−8^). PDGFB gene expression analyses of (**B**) expression quantitative trait locus (eQTL) with rs130651 and differential gene expression in samples of (**C**) AMD cases and controls and (**D**) COVID-19 stages by time from symptom onset (early ≤ 10 days and middle 11–20 days).

**Figure 3 jcm-12-00109-f003:**
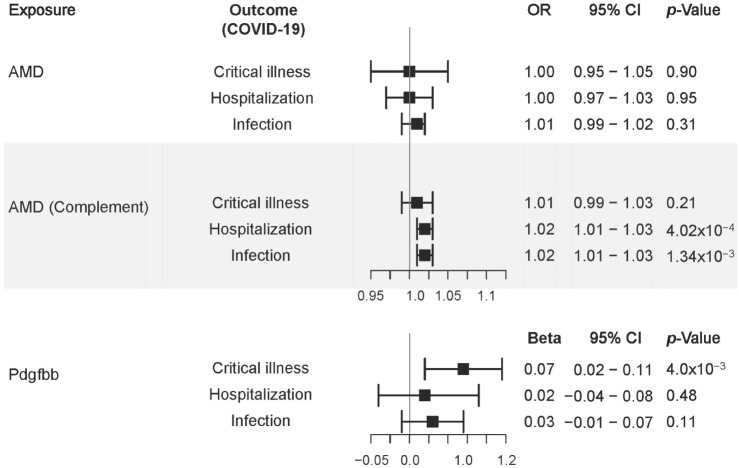
Forest plot illustrating the Mendelian randomization estimates of genetic liability of AMD risk, liability of AMD risk based on genes in the complement pathway, and human serum PDGF-BB levels (exposures) with COVID-19 phenotypes (outcomes) based on inverse-variance weighted Mendelian randomization.

**Table 1 jcm-12-00109-t001:** Genome-wide significant associations in the *PDGFB* locus identified in the MTAG analysis of AMD with three COVID-19 outcomes (critical illness, hospitalization, and infections). Rs130651 and rs4820371 are in a moderate linkage disequilibrium (LD) in the European population of 1000 genome reference (r^2^ = 0.36).

			rs130651 (EA/NEA = G/A; EAF = 0.32)			rs4820371 (EA/NEA = C/T; EAF = 0.23)		
Analytical Method	Trait		OR	95% CI	*p*	OR	95% CI	*p*
Single GWAS	AMD		0.90	0.87–0.94	1.36 × 10^−7^	0.90	0.86–0.93	1.71 × 10^−7^
	COVID-19	Critical Illness	0.96	0.89–1.04	0.27	1.01	0.93–1.09	0.78
		Hospitalization	1.00	0.96–1.04	0.87	1.00	0.94–1.06	0.93
		Infections	0.98	0.96–1.00	0.024	0.96	0.94–0.98	0.003
MTAG of AMD with COVID-19	COVID-19	Critical Illness	0.96	0.94–0.97	3.5 × 10^−8^	0.95	0.93–0.97	6.1 × 10^−8^
		Hospitalization	0.96	0.94–0.97	4.4 × 10^−8^	0.95	0.93–0.97	5.7 × 10^−8^
		Infections	0.96	0.94–0.97	2.4 × 10^−8^	0.95	0.93–0.97	2.6 × 10^−8^

EA/NEA: Effect allele/Non-effect allele. EAF: Effect allele frequency. OR: odds ratio. CI: confidence interval.

**Table 2 jcm-12-00109-t002:** Summary statistics of eQTL associations between the two SNPs (rs130651 and rs4820371) and the *PDGFB* gene expression levels from different tissue and cell types in the GTEx and FIVEx databases.

Scheme	Study	Tissue/Cell Type	Effect	SE	*p*-Value
rs130651 (*A*)	GTEx	Whole blood	0.26	0.04	1.8 × 10^−11^
	Lepik et al. 2017	Whole blood	0.31	0.031	8.5 × 10^−21^
	BLUEPRINT (Immune cells)	T cell	0.87	0.083	7.1 × 10^−20^
	DICE (Immune cells)	Th17 cell	0.96	0.081	8.7 × 10^−19^
		Tfh cell	0.63	0.056	4.7 × 10^−18^
		T cell	1.44	0.14	1.2 × 10^−15^
		Th2 cell	1.11	0.11	7.1 × 10^−15^
		CD4 T cell (naïve)	1.13	0.12	8.1 × 10^−14^
		Th1-17 cell	0.54	0.07	5.1 × 10^−11^
		T regulatory cell	0.61	0.087	9.3 × 10^−10^
		Th1 cell	0.51	0.075	3.4 × 10^−9^
		CD8 T cell (naïve)	0.85	0.13	6.2 × 10^−9^
	TwinsUK	Whole blood	0.22	0.036	3.8 × 10^−9^
rs4820371 (*T*)	GTEx	Whole blood	0.13	0.04	3.5 × 10^−3^
	Lepik et al. 2017	Whole blood	0.20	0.035	5.8 × 10^−8^

EA: effect allele. SE: standard error. PMID: literature ID in the PubMed database. GTEx: genotype-tissue expression. The eQTL results from Lepick et al. 2017, BLUEPRINT, DICE (Database of Immune Cell Expression), and TwinsUK are provided in the FIVEx database.

## Data Availability

All data generated or analyzed in this study are included in this published article.

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
