# Peer review of "Genome-Wide Pleiotropy Study Identifies Association of PDGFB with Age-Related Macular Degeneration and COVID-19 Infection Outcomes"

_jcm, 2022, doi:10.3390/jcm12010109_

Round 1
Reviewer 1 Report (Previous Reviewer 3)
The authors evaluated the genetic correlation of AMD and COVID-19 and attempted to identify genetic determinants underlying shared mechanisms of these two diseases using cross-phenotype meta-analysis. And they showed that people with AMD are an at-risk group for COVID-19 infection and severe disease, and that this risk may have a genetic basis. This is an interesting findings though the aging factor need further investigation
Reviewer 2 Report (New Reviewer)
Overall this is a well thought out , well executed and well communicated manuscript. Great paper with useful new information≥
This manuscript is a resubmission of an earlier submission. The following is a list of the peer review reports and author responses from that submission.
Round 1
Reviewer 1 Report
Thank you for the opportunity to review. An interesting manuscript. Most of the things are well explained including the limitations. Only concern:
1) Is it possible to do a subanalyses taking different subtypes of AMD? Previous studies have shown that exudative AMD might have stronger correlations with COVID 19 outcome/severity; would the correlations be stronger if only exudative AMD cases were analyzed? I understand the authors had access to only the summary statistics (IAMDGC), but it would be interesting to know the author's viewpoint on this, somewhere in the discussion.
Thank you
Author Response
We thank the reviewer for taking the time to critique our manuscript, and for their enthusiasm for the topic. We agree and understand this concern regarding the subtypes of AMD including exudative and atrophic forms. However, our analyses were conducted using AMD GWAS summary-level data, and we were unable to analyze the data based on AMD subtype. Moreover, the distribution of AMD subtypes in the GWAS dataset is highly imbalanced (67.4% exudative, 19.7% geographic atrophy, 12.9% with each form in one eye), therefore, separate analyses for each AMD subtype would not be meaningful. We have addressed this as a limitation of our study, which is now mentioned in the discussion section. In this revision, we reinforced our inherent statistical limitations in the Discussion as follows:
“Furthermore, a study of a large Korean cohort (N=135,435) showed that only the exudative (wet) form of AMD was significantly associated with severe COVID-19 illness and COVID-19 infection.6 In addition, Allegrini et al. found a significant loss in visual function among the exudative AMD patients who experienced COVID-19 infection, compared to the AMD patients before the pandemic.45 Unfortunately, we were unable to evaluate genetic correlations between COVID-19 outcomes and exudative and non-exudative AMD separately because our analyses were conducted using AMD GWAS summary-level data. Moreover, because the distribution of AMD subtypes in the GWAS dataset is highly imbalanced (67.4% exudative, 19.7% geographic atrophy, 12.9% with each form in one eye), separate analyses for each AMD subtype would not be meaningful. A balanced age distribution between the AMD and COVID-19 cohorts and AMD subtypes might have resulted in a different estimate of the genetic correlation between the two diseases."

Reviewer 2 Report
The study by Chung and colleges sought to investigate a genetic association between an AMD and COVID19 infection. Consolidating data from a variety of datasets spanning AMD and COVID-19 patients, the authors show there to be a modest correlation between the two, with the key novel finding they pose being the shared association of PDGR. Though a potential link between AMD and COVID19 patients is undoubtedly an area of interest, I do have misgiving regarding the design of the study. This would relate to the make-up of the cohorts. First, it appears according to the methods that all forms of AMD were combined when evaluating the data. I have grave concern with this given that these forms have distinct pathophysiological and molecular differences. As such the data should be analysed with these forms separated into discrete groups. Second, is the large gap in median age between groups. This does confound the findings somewhat given the well-characterised molecular effects of aging on the retina.
Author Response
We thank the reviewer for their thoughtful read and careful critiques of our manuscript. Our response to this important point is addressed in the critique response for reviewer #1 above. Our analyses were conducted using AMD GWAS summary-level data, and we were unable to analyze the data based on AMD subtype. Other databases, such as UK Biobank, also provides only summary level data on AMD without access to subtypes for analyses. We reinforced our inherent limitations in the study design. However, we think that our current study design is the best way that we could investigate the genetic relationships between the two diseases. Regarding the second point, we agree there is gap in median age between AMD and COVID19 cohorts based on the inherent clinical nature of both diseases, and this is also acknowledged as a limitation.

Reviewer 3 Report
In this manuscript, authors hypothesized that biological mechanisms shared between AMD and COVID-19 exist and attempted to identify underlying genetic factors shared between them using data from GWAS for AMD and three outcomes related to COVID-19. And they found that one novel genome-wide significant association in the PDGFB locus for AMD and COVID-19 infection. The study is well written and designed, and their conclusion is interesting and potential for further investigation.
Author Response
We appreciate giving us the opportunity to improve the quality of our manuscript. We have attempted to fix all grammatical and typographical errors in the manuscript. We apologize to the Reviewer for errors in the previous version.
